# Alternative Splicing of *TaHsfA2-7* Is Involved in the Improvement of Thermotolerance in Wheat

**DOI:** 10.3390/ijms24021014

**Published:** 2023-01-05

**Authors:** Zhenyu Ma, Mingyue Li, Huaning Zhang, Baihui Zhao, Zihui Liu, Shuonan Duan, Xiangzhao Meng, Guoliang Li, Xiulin Guo

**Affiliations:** 1Institute of Biotechnology and Food Science, Hebei Academy of Agriculture and Forestry Sciences/Plant Genetic Engineering Center of Hebei Province, Shijiazhuang 050051, China; 2College of Life Sciences, Hebei Normal University, Shijiazhuang 050024, China

**Keywords:** alternative splicing, wheat, heat stress, thermotolerance, heat shock transcription factor

## Abstract

High temperature has severely affected plant growth and development, resulting in reduced production of crops worldwide, especially wheat. Alternative splicing (AS), a crucial post-transcriptional regulatory mechanism, is involved in the growth and development of eukaryotes and the adaptation to environmental changes. Previous transcriptome data suggested that heat shock transcription factor (Hsf) *TaHsfA2-7* may form different transcripts by AS. However, it remains unclear whether this post-transcriptional regulatory mechanism of *TaHsfA2-7* is related to thermotolerance in wheat (*Triticum aestivum*). Here, we identified a novel splice variant, *TaHsfA2-7-AS*, which was induced by high temperature and played a positive role in thermotolerance regulation in wheat. Moreover, TaHsfA2-7-AS is predicted to encode a small truncated TaHsfA2-7 isoform, retaining only part of the DNA-binding domain (DBD). *TaHsfA2-7-AS* is constitutively expressed in various tissues of wheat. Notably, the expression level of *TaHsfA2-7-AS* is significantly up-regulated by heat shock (HS) during flowering and grain-filling stages in wheat. Further studies showed that TaHsfA2-7-AS was localized in the nucleus but lacked transcriptional activation activity. Ectopic expression of *TaHsfA2-7-AS* in yeast exhibited improved thermotolerance. Compared to non-transgenic plants, overexpression of *TaHsfA2-7-AS* in *Arabidopsis* results in enhanced tolerance to heat stress. Simultaneously, we also found that TaHsfA1 is directly involved in the transcriptional regulation of *TaHsfA2-7* and *TaHsfA2-7-AS*. In summary, our findings demonstrate the function of *TaHsfA2-7-AS* splicing variant in response to heat stress and establish a link between regulatory mechanisms of AS and the improvement of thermotolerance in wheat.

## 1. Introduction

High temperature caused by global warming has become one of the most significant stresses restricting crop yield and seriously threatening food security [1]. It is estimated that each degree-Celsius increase in global mean temperature would, on average, reduce global yields of wheat by 6.0%, rice by 3.2%, maize by 7.4% and soybean by 3.1% [2]. Wheat is a cold-season crop and its growth and development are often affected by high temperature, especially at flowering and grain-filling stages [3]. High temperature can induce protein misfolding and aggregation in cells, irreversible damage to membrane systems, massive destruction of microtubule organization and excessive accumulation of ROS [4]. Hsfs are critical regulators of plants in response to heat stress and their transcriptional regulation have been extensively studied [5,6,7,8,9,10]. Previous studies have shown that the plant Hsfs are strongly influenced by splicing regulation in response to heat stress [11]. However, studies on the function of Hsfs splice variants in wheat under heat stress conditions are limited.

Due to sessile growth habits, plants cannot escape adverse stresses and have to adopt complex and elaborate translational and post-translational regulatory mechanisms in response to the constantly changing environment. AS of eukaryotic transcripts is a post-transcriptional regulatory mechanism that enables cells to produce a large diversity of proteins from a limited number of genes, thus expanding the complexity of the proteome [12,13,14,15]. In plants, alternatively spliced genes are involved in a range of biological processes, including regulation of metabolic pathways, intracellular signal transduction, response to biotic and abiotic stresses and control of flowering time and circadian clocks [16,17,18,19,20,21,22]. With the development of high-throughput sequencing (HTS) technology, the number of AS events found in higher plants has increased far beyond expectation [23]. For example, AS events are estimated to account for at least 42% and 48% of intron-containing genes in *Arabidopsis* (*Arabidopsis thaliana*) and rice (*Oryza sativa*), respectively [24,25]. In fact, given that these data were obtained from plants growing under normal conditions, AS events may occur more frequently in plant response to various stresses [26]. AS is performed and controlled by a ribonucleoprotein complex called spliceosome, which is responsible for splice site selection by binding to *cis*-regulatory elements located in exons or introns [23,27]. Among the five basic types of AS observed in plants, intron retention (IR) is the predominant AS event [12]. In many AS events, such as IR, a premature translation termination codon (PTC) is found in the spliced transcript [12]. AS of pre-mRNA (precursor mRNA) directly affects gene expression or protein function [16]. On the one hand, AS forms frame shift mutations and PTC, resulting in nonsense mediated decay (NMD) and ultimately affecting mRNA stability. On the other hand, AS alters protein structure with effects on protein activity, localization, interactions with other proteins or substrates and post-translational modifications [28,29,30,31,32].

In the natural environment, plants are continuously exposed to changing temperatures. Recent studies have suggested that alternative pre-mRNA splicing may act as a “molecular thermometer” in plants, allowing them to appropriately adjust transcript abundance in response to extreme temperatures, including high temperature [12,33]. In tomato pollen, about 7500 genes were found to exhibit heat-dependent accumulation of IR and exon skipping, including 6 Hsfs and 29 heat shock proteins (Hsps) [34]. Additionally, genome-wide analysis of AS in response to HS in wheat revealed 3576 genes, with more AS events occurring on the B sub-genome than on the A and D genomes [11]. Interestingly, heat stress produces AS in different plant species, which in turn causes plants to adapt to higher temperatures. For example, in lily (*Lilium* spp.), *LlHSFA3B* is alternatively spliced under heat stress to generate splice variant *LlHSFA3B-III*, which encodes a protein localized to the cytoplasm and nucleus with no transcriptional activity and disturbs the protein interactions between LlHSFA3A-I and LlHSFA3B-I. More importantly, overexpression of *LlHSFA3B-III* in *Arabidopsis* and *Nicotiana benthamiana* showed enhanced tolerance to salinity stress and prolonged heat treatment at 40 °C, but decreased tolerance to acute HS at 45 °C [35]. In addition, the unfolded protein response (UPR) triggered by HS promotes AS of bZIP60, which is mediated by the endoplasmic reticulum (ER) localized RNA splicing factor inositol requirement enzyme 1 (IRE1). The truncated bZIP60 isoform, lacking the transmembrane domain, translocates to the nucleus and activates the expression of downstream stress-responsive genes, thereby improving thermotolerance in *Arabidopsis* [36]. Notably, HS induced AS occurred not only in transcripts of transcription factors, but also in transcripts of other HS inducible genes, such as Hsps [12]. Previous studies have shown that transcripts of Hsp genes such as *Hsp21*, *Hsp101*, *Hsp70.10*, *Hsp70.6*, *Hsp90.5* and *Hsp100.3* undergo constitutive splicing when subjected to heat priming, which would allow the plants to withstand the second stress [37]. Therefore, heat-induced AS may play an important role in the thermo-memory response of plants [12].

As so far, although the phenomenon of heat stress triggering AS events has been detected by high-throughput sequencing technology, there is little experimental evidence to show the regulatory mechanism and function of splice variants in response to high temperature and the relationship between thermotolerance and AS in wheat is also largely unknown. Based on our previous transcriptome data, we found that transcripts of *TaHsfA2-7 * may be spliced under heat stress. In this study, we further determined that *TaHsfA2-7* formed two splice variants through AS, one of which, *TaHsfA2-7-AS* could be induced by high temperature and played an important role in the improvement of thermotolerance in plants.

## 2. Results

### 2.1. Characteristics of a New Splice Variant of TaHsfA2-7

In a search of our previous transcriptome data, an AS event was identified in the transcript of *TaHsfA2-7* [38]. The results demonstrated that *TaHsfA2-7* could indeed generate two isoforms through AS, one of which is a new splice variant, named *TaHsfA2-7-AS* here (Figure 1A). The splice variant *TaHsfA2-7-AS* containing a premature translation termination codon was generated through intron retention and it was speculated to encode a small truncated protein with 121 amino acids, containing an N-terminal region, a C-terminal truncated DBD and an additional sequence rich in hydrophobic amino acids. Based on analysis of the DBD structure of human HSF1 [39] and TaHsfA2-7-AS, the C-terminal truncated DBD of TaHsfA2-7-AS contained helixes *α*1, *α*2 and *α*3 and strands β1 and β2, but lacked strands β3 and β4 (Figure 1B). AS events induced by high temperature cause changes in the transcripts of many intron-containing genes, including transcription factors. Therefore, we examined the expression level of *TaHsfA2-7-AS* in wheat at 37 °C and 45 °C. Reverse transcription quantitative PCR (RT-qPCR) results showed that the expression of *TaHsfA2-7-AS* was induced by heat stress and reached the peak at 30 min at 37 °C and 45 °C (Figure 1C). These results indicate that *TaHsfA2-7-AS* is a new HS-induced splice variant of *TaHsfA2-7*.

### 2.2. Expression Patterns of TaHsfA2-7-AS 

The expression pattern of *TaHsfA2-7-AS* in wheat is closely related to its function, so we investigated the expression profiles of *TaHsfA2-7-AS* in different tissues at seedling and reproductive stages in wheat. As shown in Figure 2A, *TaHsfA2-7-AS* was expressed differently in all examined tissues of wheat. The highest expression was found in mature leaf, followed by that in young leaf, mature shoot, young seed, stamen, mature root, pistil, young root and young shoot, whereas the transcript of *TaHsfA2-7-AS* was almost undetectable in mature seed under normal growth conditions. As a cool season crop, wheat is most vulnerable to heat stress during the reproductive stage, which has greatly adverse effects on carbon uptake and starch synthesis, resulting in the reduction of grain yield [40]. Based on this, we further explored the induction of *TaHsfA2-7-AS* under heat stress at flowering and grain-filling stages in wheat. At the two stages of reproductive development in wheat, the expression of *TaHsfA2-7-AS* was increased significantly with HS treatment at 37 °C and reached the peak at 20 min (Figure 2B), suggesting that *TaHsfA2-7-AS* may be involved in thermotolerance regulation of wheat.

### 2.3. TaHsfA2-7-AS Is Localized to Nucleus with No Transcriptional Activity

As shown in Figure 1B, TaHsfA2-7-AS is predicted to encode a small truncated TaHsfA2-7 isoform that retains only part of the DBD, so the true localization and transcriptional activity of its protein is worth considering. Firstly, the *GFP* sequence was fused to the *TaHsfA2-7* splice variant and transiently expressed in tobacco leaf epidermal cells to determine protein subcellular localization. Compared to the whole cell distribution of free GFP, the fluorescence signal of TaHsfA2-7-AS-GFP was obviously accumulated in the nucleus (Figure 3A), indicating that TaHsfA2-7-AS is localized to nucleus. Next, we investigated the transcriptional activity of TaHsfA2-7-AS in yeast with the GAL4-based yeast system. The yeast harboring pGBKT7-P53 or pGBKT7-TaHsfA2-7 grew well on the yeast medium (SD/-Trp/-His/-Ade) and could catalyze 5-bromo-4-chloro-3-indolyl-α-D-galacto-pyranoside acid (X-α-Gal), which is consistent with our earlier report [41]. However, the yeast transformed with pGBKT7-TaHsfA2-7-AS barely grew at all on the yeast medium (SD/-Trp/-His/-Ade)-like negative control, only carrying pGBKT7 (Figure 3B). Considering that the C-terminal hydrophobic sequence of TaHsfA2-7-AS may be a negative regulatory region for its transactivation, we also constructed a deletion derivatives of TaHsfA2-7-AS (D-TaHsfA2-7-AS) and analyzed its transcriptional activity in yeast. Unexpectedly, deletion of the C-terminal hydrophobic sequence of TaHsfA2-7-AS did not allow the transformed yeast to grow on yeast synthetic drop-out medium (Appendix A).

### 2.4. Ectopic Expression of TaHsfA2-7-AS Improves Thermotolerance in Yeast

To confirm the role of *TaHsfA2-7-AS* in response to heat stress, a transient expression vector TaHsfA2-7-AS-pYES2 was constructed and transformed into INVSc1 yeast strain. Meanwhile, the yeast transformed with TaHsfA2-7-pYES2 was set as the positive control and the yeast transformed with empty vector was set as the negative control. Under normal conditions, all the transformed yeast strains showed similar growth status. However, after HS, the TaHsfA2-7-pYES2 and TaHsfA2-7-AS-pYES2 transgenic yeast showed improved viability compared to the pYES2 transgenic yeast (Figure 4). Although the TaHsfA2-7-AS-pYES2 transgenic yeast exhibited weaker viability compared to the TaHsfA2-7-pYES2 transgenic yeast, it could still prove that ectopic expression of *TaHsfA2-7-AS* improves thermotolerance in yeast.

### 2.5. The Overexpression of TaHsfA2-7-AS Enhanced the Tolerance of Transgenic Arabidopsis to Heat Stress

To further explore the biological functions of *TaHsfA2-7-AS* in response to heat stress, three transgenic lines overexpressing *TaHsfA2-7-AS* in Col-0 (*3-14*, *7-23* and *12-9*) were obtained and the protein accumulations in transgenic lines were detected by western blotting (Appendix A). As shown in Figure 5A and Appendix A, under normal growth conditions, there was no significant difference in phenotype between transgenic lines and wild type (WT). However, under HS conditions, transgenic plants had fewer withered leaves and better growth status than WT. Consistent with this, the content of chlorophyll in transgenic plants was also significantly higher than that in WT, indicating a lesser degree of injury in transgenic lines after HS treatment (Figure 5B). Meanwhile, we also detected the activities of superoxide dismutase (SOD) and peroxidase (POD) in transgenic lines and WT under HS treatment or not. The results showed that under normal growth conditions, there was no remarkable difference between the transgenic lines and WT. But, when subjected to HS treatment, the activities of POD and SOD were significantly higher in transgenic lines (Figure 5C,D).

### 2.6. TaHsfA1 Acts as an Upstream Transcriptional Regulator of TaHsfA2-7 and TaHsfA2-7-AS

In plants, HsfA1s appear to be critical “master regulators” in response to HS, activating the expression of numerous genes, including *HsfA2*, *HsfA7s*, *HsfBs*, which in turn regulate the synthesis of molecular chaperones and functional enzymes [42,43,44,45]. To confirm whether the TaHsfA1 protein binds to the cis-acting heat shock element (HSE) sequence in the *TaHsfA2-7* promoter and regulates the abundance of its pre-mRNA, the fragment of *TaHsfA2-7* promoter was analyzed. Three putative *cis*-acting HSEs, named HSE1, HSE2 and HSE3, were identified in the fragment of *TaHsfA2-7* promoter (Figure 6A). Subsequently, we performed an electrophoretic mobility shift assay (EMSA) and demonstrated that the TaHsfA1 protein could bind HSE1 and HSE2 in the *TaHsfA2-7* promoter, but not HSE3 (Figure 6B). These results indicate that TaHsfA1 is directly involved in transcriptional regulation of *TaHsfA2-7* and *TaHsfA2-7-AS* by binding to HSEs located in the *TaHsfA2-7* promoter.

## 3. Discussion

### 3.1. TaHsfA2-7-AS Is a HS-Induced Splice Variant of TaHsfA2-7

The growth and development of plants are closely related to the temperature of the surrounding environment and extreme temperatures including high temperature will seriously threaten the survival of plants. To avoid or reduce the harmful effects of high temperature, plants rely on mechanisms established during evolution, such as AS, to cope effectively and ensure survival and reproductive growth [12]. In the present study, we identified a new splice variant of *TaHsfA2-7*, named *TaHsfA2-7-AS*, which was found to be induced by heat stress (Figure 1). Interestingly, Hsfs are strongly influenced by splicing regulation in responses to heat stress and the heat-induced AS regulation in plants may be an evolutionarily conserved phenomenon [46]. For example, under HS treatment, *LlHSFA3B* is alternatively spliced to produce the heat-inducible splice variant *LlHSFA3B-III* [35]. In *Arabidopsis*, various Hsfs including *HsfA2*, *HsfA7b*, *HsfB1*, *HsfB2a* and *Hsf4c*, generate intron-retained splice variants via AS in response to heat stress [47]. In addition, when rice suffers heat stress, *OsHSFA2d* is alternatively spliced into a transcriptionally active form, *OsHSFA2dI*, which participates in the heat stress response [48].

Like most splice variants, *TaHsfA2-7-AS* is also obtained through IR, which leads to the introduction of PTC (Figure 1). Although the small truncated protein encoded by *TaHsfA2-7-AS* lacks the C-terminal nuclear localization signal (NLS), it is still localized in the nucleus (Figure 3A). The small protein of TaHsfA2-7-AS may enter the nucleus through the help of other proteins containing the NLSs, but this requires further experimental validation. Coincidentally, HsfA2-III (also named S-HsfA2), a third splice variant of *HsfA2 * in *Arabidopsis*, was observed mainly in the nucleus [47]. This suggests that NLS is not the key factor for nuclear localization of Hsfs. However, there are some differences in transcriptional activation activity between TaHsfA2-7-AS and its ortholog from *Arabidopsis*. Compared with Leu-rich hydrophobic motif (LRM) in S-HsfA2, the C-terminal hydrophobic structure of TaHsfA2-7-AS had no such inhibitory effect on transcriptional activation (Appendix A).

### 3.2. TaHsfA2-7-AS Enhances Tolerance to Heat Stress

The mechanism of pre-mRNA splicing and the process of splice site selection are fairly well understood and recent efforts have been directed toward the regulatory mechanism and function of splice variants [12,13,17,23]. For example, *LlHSFA3B-III*, a heat-induced splice variant of *LlHSFA3B*, has shown considerable tolerance to both salinity stress and prolonged heat treatment at 40 °C in transgenic *Arabidopsis* and *N. benthamiana* plants [35]. Similarly, TaHsfA2-7-AS also improves thermotolerance of transgenic yeast and *Arabidopsis* in our study (Figure 4 and Figure 5). This suggested that *TaHsfA2-7-AS* can improve the thermotolerance of yeast and *Arabidopsis*. However, it is unclear whether TaHsfA2-7-AS regulates the expression of Hsp genes such as TaHsfA2-7. Indeed, when exposed to high temperature, transgenic yeast showed higher survival rates and transgenic plants exhibited better growth status, with fewer withered leaves (Figure 5A). Furthermore, the higher chlorophyll content and stronger activities of POD and SOD were observed in transgenic plants (Figure 5B–D), which also indicated that TaHsfA2-7-AS plays a positive regulatory role in response to heat stress. More importantly, the expression level of *TaHsfA2-7-AS* is dramatically induced by heat stress at flowering and grain-filling stages in wheat (Figure 2B). Therefore, it is very promising to improve the thermotolerance of wheat during the reproductive stage by overexpressing *TaHsfA2-7-AS*, so as to reduce the loss of yield.

### 3.3. The Transcription of TaHsfA2-7 and TaHSFA2-7-AS Is Regulated by TaHsfA1

To cope with heat stress, plants have developed large Hsf families, which can be roughly classified into three classes (A, B and C) based on the sequence length between DBD and HR-A/B regions and the number of amino acid residues inserted into HR-A/B region [6]. Among them, HsfA1s are the master regulators of the heat stress response in plants [42]. Transcriptomic analysis showed that more than 65% of heat-inducible genes were dependent on regulation of HsfA1s [45]. Upon heat stress, HsfA1s trigger a transcriptional cascade that is composed of many transcription factors, including *HsfA2*, *HsfA7s*, *HsfBs*. Although it has been reported that HsfA1s could induce expression of *HsfA2* in response to heat stress, related studies in wheat are limited due to the complexity of wheat genome [38,49]. In our study, we found TaHsfA1 can specifically bind to the cis-acting HSEs (HSE1 and HSE2) in the promoter sequence of *TaHsfA2-7*, thereby increasing the abundance of its pre-mRNA (Figure 6). Of course, the above conclusions need to be further investigated in wheat to explore the regulatory network of TaHsfA1-TaHsfA2-7 in response to heat stress and the relationship between thermotolerance and mechanisms of AS.

## 4. Materials and Methods

### 4.1. Plant Materials and Growth Conditions

A semi-winter heat-tolerant wheat cultivar, Cang6005, used for gene cloning and expression analysis was cultivated in Hoagland nutrient solution or soil. In addition, Columbia-0 (*Arabidopsis thaliana*) used for genetic transformation and tobacco (*Nicotiana benthamiana*) used for transient transfection were cultivated in soil. All the plants mentioned above were grown in a greenhouse under day/night conditions of 16 h/8 h (70–100 μmol·m^−2^·S^−1^), 25 °C and 50–60% relative humidity.

### 4.2. Isolation of Total RNA and Identification of Splice Variants

Total RNA was extracted from leaves of wheat using the RNArose Reagent Systems Kit (Huashun Bioengineering Co., LTD., Shanghai, China), according to the manufacturer’s protocol. RNA was further purified by using the PrimeScript^TM^ RT Reagent Kit with gDNA Eraser (TaKaRa, Dalian, China) to remove genomic DNA contamination. Then, 2 µg of purified RNA was used as template to synthesize first-strand cDNA with an oligo (dT) primer. To detect splicing variants of *TaHsfA2-7*, semi-quantitative RT-PCR was performed with specific primers designed by DNAMAN 8.0 (www.lynnon.com, accessed on 20 October 2022) software. The information of primers is presented in Appendix A.

### 4.3. Expression Analysis Using RT-qPCR

To study the expression patterns of *TaHsfA2-7-AS* in response to high temperature, seedlings of wheat at the three-leaf stage were treated with HS (37 °C and 45 °C) for different time. The second leaves were collected and then immediately frozen in liquid nitrogen for RNA extraction. To explore the tissue expression pattern of *TaHsfA2-7-AS*, the mature leaf, young leaf, mature shoot, young shoot, mature seed, young seed, mature root, young root, stamen and pistil of wheat were harvested at different growth and development stages. RT-qPCR was performed with SYBR Green Realtime PCR Master Mix (TaKaRa, Dalian, China) on a 7500realtime PCR detector (Applied Biosystems Incorporation, Foster City, CA, USA). Based on our previous methods [41], *TaRP15* [50] was selected as internal reference gene in wheat. The average values of 2^−△ct^ were used to calculate the relative expression of genes. The primers used are listed in Appendix A. Three biological replicates were performed for each qRT-PCR reaction and the results were reported as means and standard deviation. SPSS 19.0 software was used to analyze the significance.

### 4.4. Subcellular Localization

Transient expression assays were conducted as described previously [41]. The coding sequence of *TaHsfA2-7-AS* without a stop codon was inserted into the pJIT1-hGFP vector, which contains a CaMV 35S promoter and a C-terminal GFP. The reconstructed plasmid and empty plasmid were transformed into *Agrobacterium* strain GV3101 and injected into *N. benthamiana* leaves with a needleless 1 mL syringe. Before imaging, the nucleus was stained with 4′, 6-diamidino-2-phenylindole (DAPI, 1 mg·mL^−1^) dye. Fluorescence observation in the transformed *N. benthamiana* was performed with a confocal laser scanning microscope (META510, Zeiss, Jena, Germany).

### 4.5. Transactivation Analysis in Yeast

The yeast system was used for transcriptional activation activity assays. The coding sequences of *TaHsfA2-7*, *TaHsfA2-7-AS* and *D-TaHsfA2-7-AS* were cloned into pGBKT7 to fuse to GAL4 BD and then introduced into yeast strain AH109 by lithium acetate-polyethylene glycol-mediated transformation procedure, respectively. The transformants were grown on the yeast synthetic drop-out medium (SD/-Trp and SD/-Trp/-His/-Ade) for three days before observation. The transcriptional activation activities was evaluated according to their growth status and the activity of α-galactosidase. The experiment referred to the method of Titz et al. [51] and are slightly improved. The transcriptional activation was measured by *His3* gene expression and activity of α-galactosidase. The expression of His3 was observed by growing yeast cells on selective media lacking tryptophane (Trp), histidine (His) and adenine (Ade). The activity of α-galactosidase was evaluated by the depth of the color display.

### 4.6. Thermotolerance Assay in Yeast

For thermotolerance assay in yeast, the coding sequences of *TaHsfA2-7* and *TaHsfA2-7-AS* were inserted into yeast expression vector pYES2 driven by a galactose-inducible (GAL1) promoter, respectively. The reconstructed plasmids and empty plasmid were then transformed into yeast strain INVSc1 using lithium acetate-polyethylene glycol-mediated transformation procedure. The positive transgenic clones were screened and cultivated in yeast synthetic drop-out medium (SD/-Ura) with 2% (*w/v*) glucose at 30 °C for 12 h. Next, the yeast cells were resuspended and diluted to an OD600 value of 0.4 using yeast synthetic drop-out medium (SD/-Ura) with 2% (*w/v*) galactose. Before HS, the yeast cell densities were recalculated and unified in 100 µL of sterile water. Finally, after treatment for 30 min under normal conditions (30 °C) or HS conditions (50 °C), the serial dilutions were dotted on SD/-Ura plates supplemented with 2% (*w/v*) galactose to evaluate the growth status and relative survival rates. This experiment was repeated for three assays and the result of each assay was similar.

### 4.7. Generation of Transgenic Plants

To generate the overexpression construct, the coding region of *TaHsfA2-7-AS* was amplified by PCR and inserted into pCAMBIA1300 with a CaMV 35S promoter and a C-terminal GFP. The recombinant plasmid was transformed into *Agrobacterium* strain GV3101. Genetic transformation was performed using classical floral dip method at the stage of flower-budding in *Arabidopsis* [52]. The T-DNA insertion transgenic seeds were selected on MS-agar plates supplemented with 50 µg·µL^−1^ of hygromycin and confirmed by PCR. Homozygous T3 transgenic seeds were used for HS treatment.

### 4.8. Protein Extraction and Western Blotting

The seedlings were ground in liquid nitrogen and thawed in lysis buffer (50 mM Tris–HCl (pH 7.4), 150 mM NaCl, 2 mM MgCl_2_, 20% (*v/v*) glycerol, 5 mM DTT, 0.1% (*v/v*) Nonidet P-40 and 1 mM phenylmethyl-sulfonyl fluoride). The extract was centrifuged at 10,000× *g* for 5 min. The supernatant fraction was mixed with 5×SDS-PAGE loading buffer and boiled for 10 min. Protein extracts were then separated by sodium dodecyl sulphate–polyacrylamide gel electrophoresis (SDS-PAGE) and electroblotted onto polyvinylidene fluoride (PVDF) membranes (Millipore, Burlington, MA, USA). Protein detection was carried out by using anti-GFP antibody (1:5000, Sigma, Livonia, MI, USA).

### 4.9. HS Treatment in Arabidopsis

Sterilized wild-type and transgenic seeds were planted on MS agar plates. Seedlings five days old were subjected to basal thermotolerance (45 °C for 50 min, placed at 22 °C for 8 days) and acquired thermotolerance treatment (37 °C for 1 h, incubated at 22 °C for 2 days, 46 °C for 1 h, placed at 22 °C for 8 days) according to the previous methods [53]. After 8 days for recovery, photography was taken and the rosette leaves of different lines were collected to determine chlorophyll content and enzyme activity. In each experiment, about 30 seedlings of each line were treated at least and three independent experiments were conducted.

### 4.10. Measurements of Physiological Indexes

After basal thermotolerance or acquired thermotolerance treatment, the rosette leaves of *Arabidopsis* seedlings were harvested to measure the physiological indexes. Chlorophyll contents in the shoots of seedlings were measured photometrically [54]. The activities of superoxide dismutase (SOD) and peroxidase (POD) were analyzed using the nitro-tetrazolium blue chloride reduction and guaiacol colorimetric method, respectively [55].

### 4.11. EMSA

The coding sequences of TaHsfA1 was cloned into the pGEX-4T-1 vector to generate GST-TaHsfA1 fusion proteins. The expression of GST-TaHsfA1 proteins in *Escherichia coli* BL21 cells was induced by isopropyl β-D-1-thiogalactopyranoside (IPTG). The accumulation of GST-TaHsfA1 proteins purified by glutathione Sepharose 4B (GE Healthcare) was detected by SDS–PAGE. The concentrations of GST-TaHsfA1 proteins were determined using a Nano Drop 2000 spectrophotometer (Thermo Fisher, Waltham, MA, USA). The EMSA probes were labeled with biotin at the 5′-end and synthesized by Shanghai Shenggong Bioengineering Technology Co., LTD. A 10-, 100- and 1000-fold molar excess of an unlabeled DNA fragment was added to test the specificity of the GST-TaHsfA1 to the binding motif. Detection of the biotin-labeled DNA was performed using a LightShift Chemiluminescent EMSA Kit according to the manufacturer’s instructions (Thermo Scientific, Waltham, MA, USA).

## 5. Conclusions

In summary, we found that the pre-mRNA of *TaHsfA2-7* underwent AS to generate a heat-induced new splice variant, *TaHsfA2-7-AS*. Further studies showed that overexpression of *TaHsfA2-7-AS* improved the thermotolerance in transgenic yeast and *Arabidopsis*. Meanwhile, our study also demonstrated the possibility that the transcription of *TaHsfA2-7* and *TaHSFA2-7-AS* is regulated by TaHsfA1 in wheat. Therefore, our study not only elevates the understanding of the relationship between thermotolerance and AS, but also provides a new strategy for wheat breeding.

## Figures and Tables

**Figure 1 ijms-24-01014-f001:**
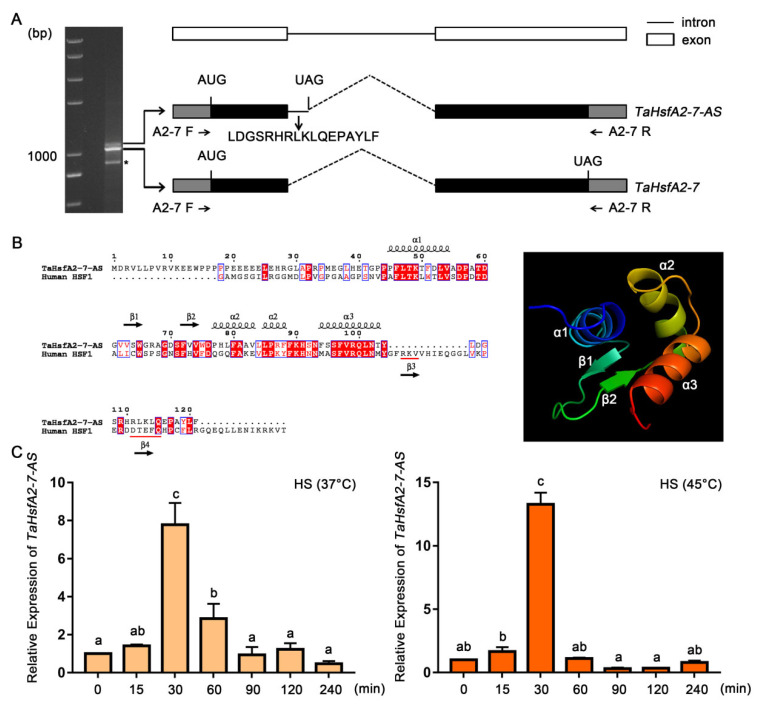
Analysis of splice variant *TaHsfA2-7-AS.* (**A**) Schematic diagram of the *TaHsfA2-7* gene and its two splice variants. Total RNA was extracted from wheat seedlings treated with high temperature (37 °C). The RT-PCR assay was performed using specific primers (A2-7 F and A2-7 R). The asterisk represents a nonspecific band. The coding sequences of two splice variants are indicated between the start codon (AUG) and the stop codon (UAG). The amino acids encoded by the retained introns are indicated by arrows. The deleted introns in two transcripts were represented by dash lines. (**B**) Sequence alignment and structure prediction of TaHsfA2-7-AS. The DBD sequence of human HSF1 (NCBI ID: 5D5U_B) was used as a template for alignment and structure prediction of TaHsfA2-7-AS with SWISS-MODEL (https://swissmodel.expasy.org/, accessed on 20 October 2022). The secondary structure elements are indicated with arrows and boxes. The PyMOL software (https://pymol.org/2/, accessed on 20 October 2022) was used to describe the 3D structure of the C-terminal truncated DBD. (**C**) Expression levels of *TaHsfA2-7-AS* in different HS conditions. Total RNA was extracted from wheat seedlings treated at 37 °C or 45 °C for the indicated time points. The expression of untreated samples was set as control. The wheat *TaRP15* gene was used as an internal control. All data represent means ± standard deviation (SD) of three biological replicates. Different lowercase letters above the bars denote significant differences at the *p* < 0.05 level.

**Figure 2 ijms-24-01014-f002:**
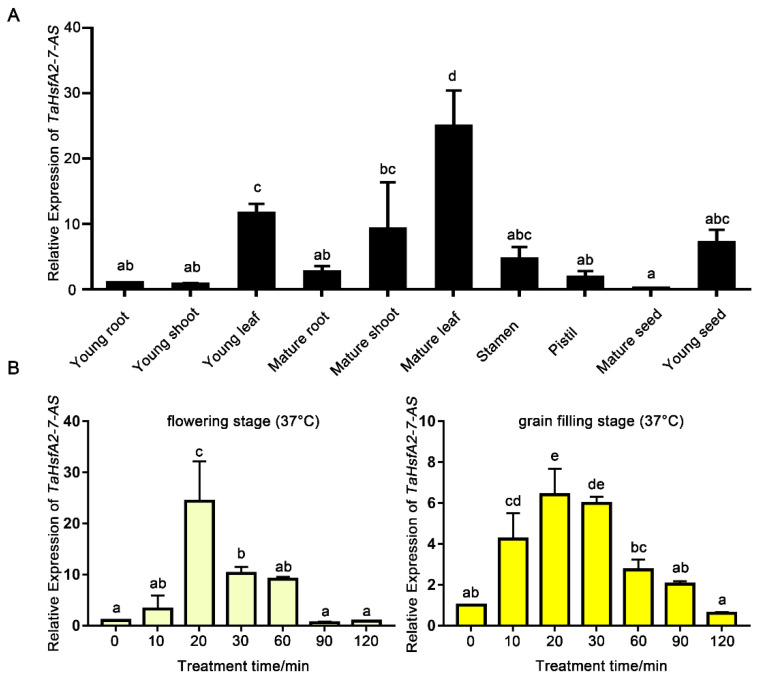
Expression analysis of *TaHsfA2-7-AS. * (**A**) Tissue-specific expression patterns of *TaHsfA2-7-AS*. RNA samples were from tissue of wheat at different growth and development stages. The expression level in young root was set as “1”. (**B**) Expression levels of *TaHsfA2-7-AS* under heat stress at flowering and grain-filling stages. Wheat at flowering and growth stages was subjected to HS treatment (37 °C), from which total RNA was extracted and analyzed by RT-qPCR. The control without HS was established in parallel. The wheat *TaRP15* gene was used as an internal control. Bars are means ± standard deviation (SD) of three biological replicates. Different lowercase letters above the bars denote significant differences at the *p* < 0.05 level.

**Figure 3 ijms-24-01014-f003:**
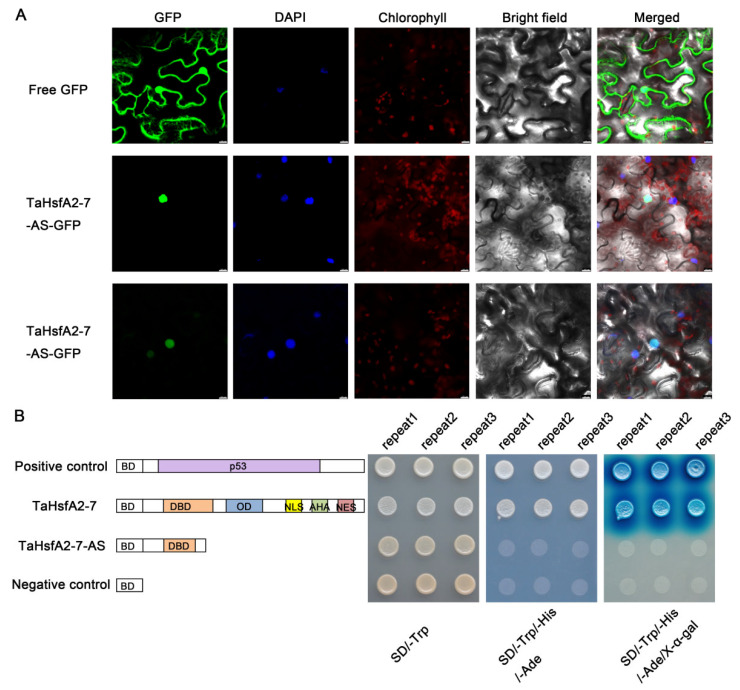
Subcellular localization and transcriptional activity analyses of TaHsfA2-7-AS. (**A**) Subcellular localization of TaHsfA2-7-AS in tobacco leaf epidermal cells. The 35S::GFP (control) and 35S::TaHsfA2-7-AS-GFP were transformed into tobacco leaf epidermal cells and visualized by DAPI staining. Bars = 10 μm. (**B**) Transcriptional activation analysis of TaHsfA2-7-AS in yeast. Fusion proteins of the GAL4 DNA-binding domain (BD) and TaHsfA2-7 or TaHsfA2-7-AS were expressed in the yeast strain AH109. The transformed yeast cells were spotted onto SD/-Trp and SD/-Trp/-His/-Ade media. The plates were incubated at 30 °C for 3 days. The pGBKT7 and pGBKT7-P53 vectors were used as negative and positive controls, respectively.

**Figure 4 ijms-24-01014-f004:**
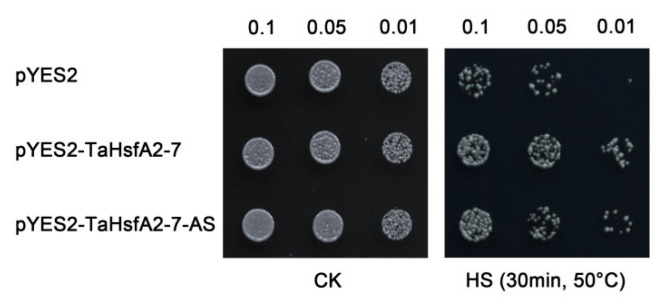
Thermotolerance assay of TaHsfA2-7-AS in yeast. Growth status of transgenic yeast are assessed in response to heat stress by spotting on SD/-Ura media. The control without HS was established in parallel. 0.1, 0.05, 0.01 represent the serial gradient dilutions.

**Figure 5 ijms-24-01014-f005:**
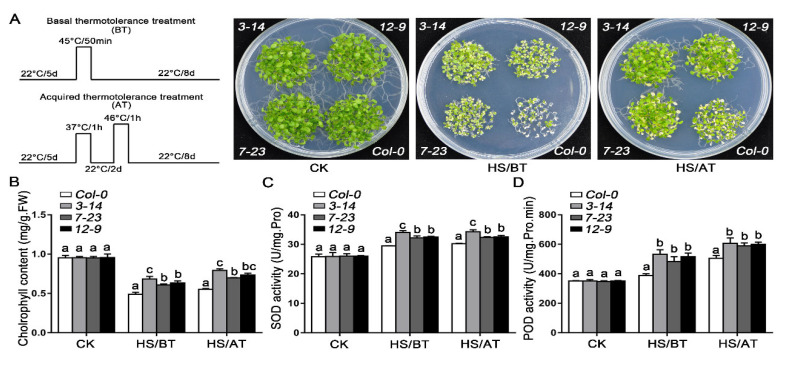
Thermotolerance assays of TaHsfA2-7-AS in *Arabidopsis*. (**A**) The phenotypes of WT and transgenic seedlings after basal thermotolerance (BT, 45 °C for 50 min, placed at 22 °C for 8 days) and acquired thermotolerance (AT, 37 °C for 1 h, incubated at 22 °C for 2 days, 46 °C for 1 h, placed at 22 °C for 8 days) treatment. The plants were photographed 8 d after different HS. (**B**) Chlorophyll content of WT and transgenic seedlings treated with different HS. (**C**,**D**) POD and SOD activities of WT and transgenic seedlings under HS treatment or not. For each experiment, at least 30 plants per line were used. Values are means ± SD from three independent measurements. Different lowercase letters above the bars denote significant differences at the *p* < 0.05 level.

**Figure 6 ijms-24-01014-f006:**
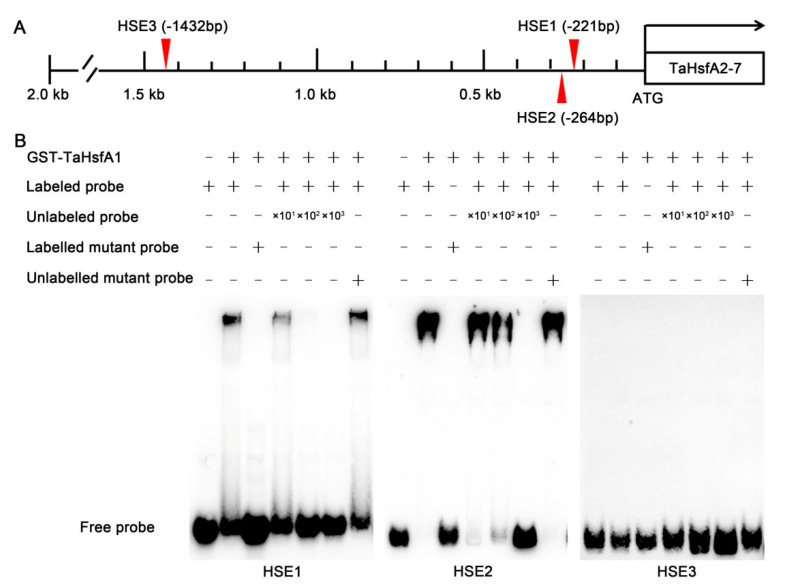
TaHsfA1 regulates the expression of *TaHsfA2-7* and *TaHSFA2-7-AS.* (**A**) Diagram of the *TaHsfA2-7* promoter. The *cis*-acting HSEs (HSE1, HSE2 and HSE3) in the promoter sequence of *TaHsfA2-7 * are marked with red triangles. (**B**) EMSA of TaHsfA1 binding to the cis-acting HSEs (HSE1, HSE2 and HSE3) in the *TaHsfA2-7* promoter. Biotin-labeled probe containing the *cis*-acting HSEs (HSE1, HSE2 and HSE3) from *TaHsfA2-7* promoter was incubated with GST-TaHsfA1 in vitro. An unlabeled probe or unlabeled mutated probe was used for competition and the biotin-labeled mutated probe was used as negative control.

## Data Availability

Not applicable.

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
