# Peer review of "Alternative Splicing of TaHsfA2-7 Is Involved in the Improvement of Thermotolerance in Wheat"

_ijms, 2023, doi:10.3390/ijms24021014_

Round 1

Reviewer 1 Report

The authors present a large body of work providing evidence that the heat shock transcription factor HSFA2 is alternatively spliced under heat stress and that the variant (AS), which has a truncated DNA binding domain, is conferring heat tolerance to yeast and Arabidopsis. The authors suggest that, based on a GFP-reporter study, that the AS version is in the nucleus despite the absence of a nuclear localization signal, but that it does not have transcriptional activation activity as can be expected. The mode of action of this short AS transcript remains unclear and it is unfortunately not discussed in sufficient detail.  

As listed below, there are many questions about the data presented the need to be addressed.   

Specific comments:

Lines 109-116: Reduce the number of acronyms used, it is very difficult to follow

Line 110/111: The sentence “To further validate…PCR was performed” can be deleted, its redundant with further below in the paragraph.

Figure 1:

1B: include the full length sequence in the alignment and also show both  protein models

It is unclear what the domains indicated in the human sequence show

I don’t understand the legend: how were Swiss Model and PyMOL were combined in the model?

1C: Since the same primer pairs were used for the qRT-PCR, how were the two products differentiated? Details on the melt curve need to be included.   

Figure 2:

As stated above, it needs to be clarified how the two PCR amplicons were differentiated. Actually, the experiment should have been conducted with variant-specific primers.

What tissue was analysed in 2B - leaves?

Details on the reference gene need to be provided demonstrating that it does it show stable expression under heat stress?

 Line 334: what is a “one-heart stage”?

Figure 3 and corresponding text:

This needs to be explained and discussed in much more detail. How can the HSFA2-AS be in the nucleus without an NLS?

Fig 3A there are five nuclei stained with DAPI but only one shows the GFP signal. I am also surprised that there is no background signal at all. How often was that experiment repeated? Multiple nuclei need to be shown to make this more believable.

Fig B: The principle of the GAL4 assay needs to be explained in detail and references provided in the methods part. I am not familiar with the assay – what exactly is interacting with what here to activate expression?  

Figure 4: I find it surprising that the wheat HSFA2 confers thermotolerance in yeast. This suggests that the HSFA2 downstream targets are conserved?  This needs to be explained in much more details. Also, it needs to be stated how many independent experiments were performed. The one image shown in Fig 4 is not convincing and does suffice unless results were replicated.

Related to that, has it been shown in wheat the HSFA2 is a positive regulator of POD and SOD? This needs to be established if the authors want to causally relate enzyme activity with HSFA2.

Figure 6: The sequences of the three HSEs need to be provided. How much flanking sequence was included?

Discussion: The discussion requires more depth and critical assessment of the data. As is, it is largely a repetition of the data and literature review.

M&M: Overall the methods need to be explained in more detail and references need to be provided for all methods and equipment used.

The number of reps need to be clearly stated for all experiments

Author Response

Dear Dr. Reviewer,

Thank you very much for reading and commenting on our paper “Alternative Splicing of TaHsfA2-7 Is Involved in the Improvement of Thermotolerance in Wheat”. We revised our manuscript based on the comments. The following is an itemized list of all changes and explanation in response to the comments point by point.

We redued the number of acronyms from line 109 to line 116.

The reduntant sentence “to further … PCR was performed” (Line 110/111) had been deleted.

Figure1:

Figure 1B was redrawn and include the full length sequence in the alignment. The protein model of human HSF1 was published before in the paper (Neudegger et al., 2016, Nature Structural & Molecular Biology) and we added the reference in the revised manuscript.

We labeled clearly the domains indicated in the sequence of human HSF1.

SWISS-MODEL was used for alignment and structure prediction of proteins, but PyMOL was used to optimize the image of the 3D structure of TaHsfA2-7-AS and make the image clearer.

In Figure 1C, HS at 37°C and HS at 45°C are two separate experiments. Although the same primer pairs were used for the qRT-PCR, it is not required to differentiate the two products.

Figure 2:

In Figure 2B, flowering stage at 37°C and grain filling stage at 37°C are two separate experiments, it is not required to differentiate the two products.

Leaves were analysed in Figure 2B and we explained in the legend.

We added the reference paper (Xue et al., 2014, Journal of Experimental Botany) about details on the reference gene in the revised manuscript.

We modified the expression of the sentence in Line 334.

Figure 3 and corresponding text:

We explained and discussed in much more detail about the HSFA2-AS located in the nucleus without an NLS.

We repeated the experiment twice about Figure 3A. And we provided new photographs to show multiple nuclei.

We added the principle of the GAL4 assay in the methods part and provided the reference. The assay is a typical experiment for transcriptional activation activity analysis.

Figure 4:

We added explanation that the wheat HSFA2 confers thermotolerance in yeast in discussion part. We stated that three independent experiments were performed in the related methods about thermotolerance assay in yeast.

POD and SOD are the typical physiological indicators of evaluating plant thermotolerance. The two physiological indicators were used in many papers. Examples are as follows: Ru et al. (2022, Plant Science), Meng et al. (2022, Frontiers in Plant Science), Guo et al. (2022, BMC Plant Biology).

Figure 6:

We provided the probe sequences of the three HSEs in supplementary data Table S1. The left flanking sequence of HSEs was 15 bp. The right flanking sequence of HSE1 and HSE2 was 17 bp and the right flanking sequence of HSE1 was 20 bp.

Discussion:

We added more depth and critical assessment of the data. Please check it in the revised manuscript.

We added details and references need to be provided for all methods and equipment used.

We provided the number of repeats for all biological repeat experiments.

Thank you!

Guoliang Li

Institute of Biotechnology and Food Science,

Hebei Academy of Agriculture and Forestry Sciences,

598 West Heping Road,

Shijiazhuang 050051

China

Reviewer 2 Report

High temperature affects plant growth and development, resulting in reduced production of crops worldwide, especially wheat. As a crucial post-transcriptional regulatory even, Alternative splicing (AS) is involved in the growth and development of eukaryotes and the adaptation to environmental changes. Authors identified a novel splice variant, TaHsfA2-7-AS, was induced by high temperature and played a positive role in thermotolerance regulation in wheat by previous transcriptome data analysis. Which is predicted to encode a small truncated TaHsfA2-7 isoform, retaining only part of the DNA-binding domain (DBD), and was localized in the nucleus. The expression level of TaHsfA2-7-AS is significantly up-regulated by heat shock (HS) during flowering and grain-filling stages in wheat but lacked transcriptional activation activity. By ectopic expression of TaHsfA2-7-AS in yeast and Arabidopsis exhibited improved thermotolerance for thermo-stress. TaHsfA1 may directly involve in the transcriptional regulation of TaHsfA2-7 and TaHsfA2-7-AS. Their findings demonstrate the function of TaHsfA2-7 splicing variant in response to heat stress and establish a link between regulatory mechanisms of AS and the improvement of thermotolerance in wheat. The manuscript is of high quality and written in a consistent manner. The research has appropriate methodology, reasonable design, and good results. Which have critical worth and reference importance for the investigation of wheat's heat resistance, but there are still a few minor modifications that need to be added, such as the problem with Latin italics in the literature and the expression of TaHsfA1. Suggesting that it is recommended to publish in this journal after revising.

Author Response

Dear Dr. Reviewer,

Thank you very much for reading and commenting on our paper “Alternative Splicing of TaHsfA2-7 Is Involved in the Improvement of Thermotolerance in Wheat”. We added a few minor modifications, such as the problem with Latin italics in the literature and the expression of TaHsfA1. Please check them in the revised manuscript.

Thank you again!

Guoliang Li

Institute of Biotechnology and Food Science,

Hebei Academy of Agriculture and Forestry Sciences,

598 West Heping Road,

Shijiazhuang 050051

China